Light enhanced calcification in Stylophora pistillata: effects of glucose, glycerol and oxygen

Holcomb Michael 1 mholcomb3051@gmail.com
Tambutté Eric
Allemand Denis
Tambutté Sylvie
Centre Scientifique de Monaco , Monaco
Medina Mónica
1 Current affiliation: ARC Centre of Excellence in Coral Reef Studies, School of Earth and Environment & Oceans Institute, The University of Western Australia, Crawley, WA, Australia

Electronic publication date: 2014 May 13
Publication date: 2014
Volume: 2
Electronic Location ID: e375
Received 2013 Dec 30; Accepted 2014 Apr 19
Copyright: © 2014 Holcomb et al.
Copyright year: 2014
Copyright holder: Holcomb et al.
License: This is an open access article distributed under the terms of the Creative Commons Attribution License, which permits unrestricted use, distribution, and reproduction in any medium, provided the original author and source are credited.
License URL: https://creativecommons.org/licenses/by/3.0/

Keywords: Coral, Oxygen, Calcification, Respiration, Fixed carbon

Funding: Government of the Principality of Monaco National Science Foundation International Research Fellowship The Centre Scientifique de Monaco is funded by the Government of the Principality of Monaco. M Holcomb was supported by a National Science Foundation International Research Fellowship. The funders had no role in study design, data collection and analysis, decision to publish, or preparation of the manuscript.

==============================
Zooxanthellate corals have long been known to calcify faster in the light than in the dark, however the mechanism underlying this process has been uncertain. Here we tested the effects of oxygen under controlled pCO2 conditions and fixed carbon sources on calcification in zooxanthellate and bleached microcolonies of the branching coral Stylophora pistillata. In zooxanthellate microcolonies, oxygen increased dark calcification rates to levels comparable to those measured in the light. However in bleached microcolonies oxygen alone did not enhance calcification, but when combined with a fixed carbon source (glucose or glycerol), calcification increased. Respiration rates increased in response to oxygen with greater increases when oxygen is combined with fixed carbon. ATP content was largely unaffected by treatments, with the exception of glycerol which decreased ATP levels.

Introduction

Rising ocean temperatures and declining pH have received considerable attention in recent years (e.g., Hoegh-Guldberg et al., 2007; McCulloch et al., 2012). Among the major concerns are the impacts on coral reefs, particularly on corals themselves. As temperatures rise, bleaching events (the loss of symbiotic zooxanthellae) become more frequent and severe, reducing the ability of the coral to cope with other stressors, leading to the loss of coral reefs (e.g., McWilliams et al., 2005). Rising CO2 levels further compound the problem by reducing aragonite saturation state and potentially calcification (e.g., Langdon & Atkinson, 2005; Holcomb, McCorkle & Cohen, 2010).

Central to our understanding of the response of corals to changing environmental conditions is an understanding of the role the symbiotic zooxanthellae play in coral calcification. It has long been established that calcification rates increase in zooxanthellate corals during periods in which photosynthesis is occurring (Yonge, 1931), a process known as light-enhanced calcification (e.g., Vandermeulen, Davis & Muscatine, 1972; Gattuso, Allemand & Frankignoulle, 1999). Comparisons of facultatively symbiotic corals with and without zooxanthellae similarly show increased calcification in the presence of symbionts (Jacques, Marshall & Pilson, 1983). Yet, comparisons of zooxanthellate corals to azooxanthellate corals show that some azooxanthellate corals can grow just as rapidly (Marshall, 1996; Mortensen & Rapp, 1998), bleached corals can continue calcifying for some time at rates similar to those of unbleached corals (Rodrigues & Grottoli, 2006), in corals such as acroporids, the fastest growing regions are largely azooxanthellate (Pearse & Muscatine, 1971), and light does not necessarily stimulate calcification in zooxanthellate corals (e.g., Rinkevich & Loya, 1984). Thus, it is not clear that symbionts per se enhance calcification.

There are several mechanisms which could be responsible for the increase in calcification generally associated with symbiosis (for reviews see: Gattuso, Allemand & Frankignoulle, 1999; Allemand et al., 2011; Tambutté et al., 2011). Calcification could increase due to the drawdown in CO2 as a direct result of photosynthesis, leading to increased pH, shifting the dominate CO2 species from bicarbonate to carbonate and increasing the saturation state (Kawaguti & Sakumoto, 1948). Other likely mechanisms by which calcification could be increased include the use of fixed carbon from photosynthesis as fuel to increase ion pumping and elevate saturation state (Goreau, 1959), or the use of fixed carbon for organic matrix synthesis, which may facilitate aragonite precipitation (Young, O’Connor & Muscatine, 1971). Oxygen produced by the zooxanthellae may relieve internal hypoxia and thus enhance calcification (Rinkevich & Loya, 1984; Rands et al., 1992; Nakamura, Nadaoka & Watanabe, 2013). Or symbionts may remove compounds which inhibit calcification or metabolism (e.g., Goreau, 1961; Vandermeulen & Muscatine, 1974).

Though numerous studies have examined the production of fixed carbon by zooxanthellae and suggested that various forms of fixed carbon (glycerol (Muscatine, 1967), glucose (Burriesci, Raab & Pringle, 2012), among other compounds (e.g., Whitehead & Douglas, 2003)) are translocated to the coral host, the role of fixed carbon specifically in calcification has received little attention. Various carbon compounds are known to be taken up by corals. Stephens (1960), Stephens (1962), and Lewis & Smith (1971) demonstrated that corals have the ability to take up fixed carbon compounds supplied exogenously. Stephens (1962) suggested that a glucose concentration of ∼80 µM is sufficient for glucose uptake to balance respiration in Fungia. Oku & Yamashiro (2003) and Oku, Yamashiro & Onaga (2003) claimed corals have the ability to incorporate a range of exogenous sugars into their tissues, and that supplying sugars exogenously can improve the survival of bleached corals. Thus, exogenous fixed carbon may substitute for that provided by zooxanthellae.

The effect of fixed carbon supplementation on calcification has been examined in five studies. Taylor (1977) showed that fixed carbon compounds supplied exogenously are transported to the regions of most rapid growth. Vandermeulen & Muscatine (1974) and Chalker (1975) exposed corals in the light and in the dark to glucose, glycerol and alanine. Calcification rates were either unchanged or declined as a result of exposure to fixed carbon sources at concentrations from 10 µM to 10 mM. A similar result was found by Al-Horani, Tambutté & Allemand (2007) for both glucose and feeding with Artemia. In contrast, Colombo-Pallotta, Rodríguez-Román & Iglesias-Prieto (2010) found that in bleached corals, glycerol could stimulate calcification, though not in zooxanthellate corals. Such results suggest that fixed carbon by itself is not generally limiting the calcification process.

In addition to fixed carbon, zooxanthellae also provide oxygen to the coral host. Several studies have documented the daily cycle in dissolved oxygen levels adjacent to the coral tissue, with hyperoxia persisting for much of the day, and hypoxia persisting for much of the night (e.g., Rands et al., 1992; Shashar, Cohen & Loya, 1993; Marshall & Clode, 2003; Agostini et al., 2011). Respiration and calcification rates are linked to this cycle in oxygen saturation, with lower rates at reduced oxygen levels (Rinkevich & Loya, 1984; Shick, 1990; Al-Horani, Tambutté & Allemand, 2007) and increased calcification rates under elevated oxygen (Colombo-Pallotta, Rodríguez-Román & Iglesias-Prieto, 2010; Wijgerde et al., 2012; Nakamura, Nadaoka & Watanabe, 2013). In both the studies of Vandermeulen & Muscatine (1974) and Chalker (1975), corals were incubated in unstirred, un-aerated containers, thus any potential benefit from the supplied carbon sources may have been masked by oxygen limitation. Under normal conditions, photosynthate is supplied and oxygen levels are elevated by photosynthesis, thus it may be the combination of oxygen and fixed carbon that allow the coral to respire and calcify faster.

A lack of, or insufficient, flow may be an important factor limiting the potential benefits of fixed carbon in early experiments. Corals, like many benthic organisms, depend upon water movement for their supply of nutrients, food, and waste removal. Water flow plays a major role in determining the diffusive boundary layers surrounding corals, affecting transport rates of O2, CO2, nutrients, etc. likely affecting all aspects of coral physiology (Dennison & Barnes, 1988; Atkinson & Bilger, 1992; Kuhl et al., 1995; Nakamura & Van Woesik, 2001). Respiration in particular may be limited by the diffusion of oxygen under stagnant or low-flow conditions (Shick, 1990; Kuhl et al., 1995), which would in-turn limit energy availability to other processes, such as calcification. Thus adequate flow is likely critical if any benefit is to be seen to supplementation with fixed carbon.

Given that respiration and calcification rates respond to elevated oxygen levels, dark calcification may indeed be energy limited due to insufficient oxygen to maintain oxidative phosphorylation. Fixed carbon availability may also limit calcification if sufficient oxygen is available. Thus, elevated oxygen levels in combination with fixed carbon supplementation may be sufficient to increase dark calcification rates to levels comparable to light calcification rates. The primary energy carrier used to convert energy derived from respiration into ion pumping and other energy requiring activities is adenosine triphosphate (ATP), thus energy limitation may be reflected in cellular ATP content. Here combinations of different fixed carbon sources and concentrations and elevated oxygen levels were tested on zooxanthellate and bleached microcolonies of the branching coral Stylophora pistillata to determine whether oxygen limitation or a combination of oxygen and fixed carbon limitation can explain light enhanced calcification. In addition to calcification, respiration and ATP were measured to provide insight into the potential underlying processes.

Materials & Methods

Experimental organisms

Colonies of the branching coral Stylophora pistillata were maintained in the laboratories of the Centre Scientifique de Monaco in flow-through aquaria receiving Mediterranean seawater (S = 38) heated to 25 °C. Experimental microcolonies were prepared as described by Al-Moghrabi, Allemand & Jaubert (1993) and Ferrier-Pages et al. (2003), composed of ∼2 cm × ∼1 cm terminal portions of branches on nylon monofilament line. Microcolonies were used after at least three weeks of recovery, allowing tissue to fully cover the cut skeleton. Lighting was provided by JBL Marin Day T5 bulbs providing an irradiance of ∼195 µmol photons m−2 s−1 at the position of the microcolonies with a 12 h light/dark cycle. To prepare bleached microcolonies, microcolonies were maintained in the dark for a minimum of six weeks prior to use, at which point microcolonies appeared pure white. Microcolonies were fed twice a week with newly hatched Artemia sp. A subset of microcolonies were weighed periodically to estimate growth rates (buoyant weight technique, Jokiel, Maragos & Franzisket, 1978; Davies, 1989).

Experimental incubations

Exposures to treatment conditions, calcification, and respiration rate measurements were carried out in 250 ml Erlenmeyer flasks (made of borosilicate glass). Flasks were sealed using transparent acrylic stoppers. A small hole in the center of the acrylic stopper allowed a microcolony to be suspended in the center of the flask; a rubber stopper was used to seal the hole once the microcolony had been placed level with the 200 ml mark (Fig. S1). Preliminary experiments showed oxygen influx into nitrogen sparged seawater and oxygen efflux from oxygen sparged seawater were negligible with this sealing system. Flasks were placed on a multiposition magnetic stir plate and stirred at 145 rpm with teflon coated stir bars. Different stir bar sizes (7.8 × 39.2 mm, 5.9 × 15 mm, and 4.5 × 11.95 mm—values are average measured diameter and length, the manufacturer’s stated size was generally slightly larger) were tested initially (Fig. 1), different stirring speeds were tested as well (data not shown, see Supplemental Information). Flasks were held within a constant temperature bath set at 25 °C. Incubations were carried out either in the dark or under ∼195 µmol photons m−2s−1 provided by JBL Marin Day T5 bulbs. Microcolonies were starved for at least two days prior to being used in experiments to reduce nutrient efflux. All dark incubations for zooxanthellate colonies were started in the morning such that microcolonies had been pre-conditioned to dark conditions for 12 h prior to the start of experiments. Light incubations were carried out during normally lit hours. Corals were maintained under treatment conditions for at least 20 min prior to starting the experiment to allow the coral to acclimate to treatment conditions. All seawater used for incubations was passed through two 0.45 µm membrane filters prior to use. The seawater mass present in each incubation was estimated based on the mass of seawater held by each incubation chamber without a coral and the displacement volume of the coral skeleton calculated from skeletal dry weight and assuming a density of 2.9 g/cm3. For each experimental run, in addition to treatment incubations, seawater only and treatment seawater incubations were carried out to check for background changes and corals not exposed to treatment conditions were included as well to check for day-to-day variations in growth/metabolic rates. Following incubations, water from a subset of flasks which had previously contained corals was pooled and used for an additional incubation (without coral) to assess the potential for microbial growth in the treatment seawater to impact the measured values. Dark incubations of corals generally resulted in a ∼14% decline in oxygen and a ∼4% decline in alkalinity by the end of the incubation. In some instances (for bleached microcolonies), the change in alkalinity was below the resolution of the measurement method, calcification data for such incubations were discarded.

Figure 1 Effect of stir bar size.

Effect of stir bar size on relative calcification rates (A) and oxygen consumption (B) by zooxanthellate microcolonies maintained in the light or in the dark, the 40 mm stir bar was used as the control value for calculating relative rates. N = 4 or greater for all treatments, symbols are mean, error bars are standard deviation.

Water chemistry

Oxygen

Experimental oxygen and CO2 levels were set by bubbling seawater held at 25 °C with humidified air or mixed gas, the composition of the mixed gas was set using mass flow controllers to mix ambient air, oxygen, and CO2 to achieve ∼2× ambient pO2 at ambient p CO2. CO2 levels were verified using a Licor 6262 CO2 analyzer. Oxygen (% saturation) was measured using a fiber optic oxygen measurement system (PreSens) with Pst3 sensors mounted on the inside of each flask. Sensors were calibrated with air (100%) and N2 (0%) sparged seawater, a linear response was assumed. Atmospheric pressure was recorded at the time each experiment was performed. Oxygen concentrations were calculated as follows. The oxygen concentration for air saturated (100%) seawater at the experimental salinity and pressure was calculated using data from Benson & Krause (1984) as fit by equation 8 of Garcia & Gordon (1992), the first A3 term was ignored. Concentrations calculated in this manner are for 1 atm total pressure, values were corrected for the measured atmospheric pressure and used to convert percent saturation measurements to concentrations.

Carbonate chemistry

Alkalinity samples were collected at the start and end of each experiment in pre-cleaned 20 ml scintillation vials and stored at 4 °C till measured. Semi-automated titrations with 0.03 N HCl (containing 41 g/L NaCl) were carried out on ∼4 g replicated samples using a Metrohm Titrando 808 dosimat. Certified reference material (supplied by the laboratory of Andrew Dickson, Scripps Institute of Oceanography) and an internal seawater standard were run each time samples were measured, alkalinity was calculated via regression per Holcomb, Cohen & McCorkle (2012).

Samples for pH measurement were collected in 20 ml syringes, capped, and kept in a 25 °C water bath until measured. An initial sample was injected into a 10 cm fiber optic flow cell for collecting the reference spectrum, then ∼1 µl of a 1.47 mM meta Cresol Purple (Acros 199250050 lot: A0264321) solution was added per ml seawater in the syringe using a micrometer syringe (Gilmont) to inject and mix the mCP. The solution was then injected into the flow cell and the pH measured, calculations followed DOE (1994).

Alkalinity, ammonia, and pH were then used to estimate calcification assuming calcification alters alkalinity by 2 moles per mole CaCO3 produced, ammonia was assumed to increase alkalinity by an amount equal to its concentration. Carbonate chemistry parameters were calculated using CO2Sys (van Heuven et al., 2009), additional species contributing to alkalinity were included as appropriate.

Fixed carbon

All incubations with fixed carbon substrates included antibiotics (35 IU penicillin, 0.035 mg streptomycin per ml) and antibiotics were included in the associated control incubations. The effects of this antibiotic concentration on calcification and respiration were tested as well. Glucose or glycerol was added as a concentrated stock solution. Target concentrations were ∼1% (120 mM) glycerol, 1 mM glycerol, 1 mM glucose, and 0.02 mM glucose. Concentrations were chosen based on the works of Whitehead & Douglas (2003) and Colombo-Pallotta, Rodríguez-Román & Iglesias-Prieto (2010). Concentrations of each respiratory substrate were verified at the start and end of each incubation using one of the following kits, protocols were modified slightly to improve results in seawater matrices: d-glucose kit (Megazyme) for glucose, glycerol GK kit (Megazyme) for glycerol. Measured concentrations were: 110–130 mM and 1–1.4 mM for glycerol, 0.9–1 mM and 0.019–0.021 mM for glucose. Concentrations of all fixed carbon compounds changed by less than 10% during the incubations, except for the 0.02 mM glucose treatment for which concentration declined by ∼40% by the end of the incubation, which is consistent with estimates of the uptake of fixed carbon substrates (Stephens, 1960; Whitehead & Douglas, 2003).

Nutrients

Ammonia and phosphate concentrations were measured at the start and end of a subset of incubations using a Proxima autoanalyzer (Alliance Instruments) following the manufactures recommended protocols. No nutrient release was detected for zooxanthellate microcolonies, only for bleached microcolonies were nutrient releases measureable and thus alkalinity data corrected for changes in nutrient concentrations.

ATP

ATP was extracted with sulfuric acid following a protocol modified from Fang, Chen & Soong (1987) and Fang, Chen & Chen (1989). Microcolonies were placed in 0.6 N sulfuric acid (kept on water/ice prior to use) and placed immediately in a sonicator (Branson 200 Ultrasonic Cleaner) filled with water/ice. Samples were sonicated for 15 min, tubes with microcolonies re-weighed, and a sample of the extract transferred to a 1.5 ml polypropylene tube (Eppendorf) and stored at −80 °C until measured. See Supplemental Information for a comparison of this method with other techniques.

ATP measurements were made using a luciferin/luciferase based ATP detection kit (Roche ATP Bioluminescence Assay Kit HS II) with light emission measured using either a Lumat 9507 or 9508 luminometer (Berthold), and concentrations calculated based on standard additions. A detailed protocol is included in the Supplemental Information.

Normalizations

Protein was extracted from the tissue following ATP extraction by adding NaOH to produce a ∼1 N NaOH solution which was held at 90 °C for ∼20 min. Protein was measured using a Pierce BCA protein assay kit; protein concentrations were calculated using standard addition (using BSA as the standard).

Surface area was determined via the wax method of Stimson & Kinzie (1991). Normalization data are summarized in Table S2.

Statistics

To calculate total protein and ATP per coral it was assumed that the internal water volume of the microcolony equilibrated with the surrounding solution, thus the extract volume consisted of the volume of the extraction solution added plus the volume of seawater added with the microcolony (approximated by the change in weight of the tube upon addition of the coral minus the dry weight of the coral).

ATP extracted from corals was normalized to surface area, protein, and skeletal dry weight and compared using a one-way ANOVA. Measurements of calcification and respiration under treatment conditions were normalized to measurements on the same coral under ambient (dark) conditions, to account for day to day variations in rates, all rates were normalized to untreated controls measured at the same time (per Holcomb, Cohen & McCorkle, 2012), data were subsequently compared using ANOVA. When the overall ANOVA model showed a significant treatment effect, treatments significantly different from each other were detected using Dunnett’s multiple comparison to compare treatments to controls. Residuals were plotted against predicted values and on normal probability plots to assess violations of ANOVA assumptions. The SAS software package was used for all statistical tests.

Results

Effects of stirring on physiological parameters

Several stirring rates and stir bar sizes were tested to establish appropriate flow rates for the experimental incubations (Fig. 1). Significant differences were detected in dark respiration rates among stirring treatments (F2,13 = 18.68, p = 0.0002), with smaller stir bars being associated with lower respiration rates. No significant differences in calcification were detected, though there was an apparent trend toward higher dark calcification rates with larger stir bar sizes. A stirring speed of 145 rpm with a 39.2 × 7.8 mm (manufacturers stated size: 40 × 8 mm) stir bar was chosen for the experiments since it allowed maximal respiration and calcification rates while maintaining normal polyp expansion (visually assessed, not quantified).

Effect of treatments on calcification

Dark calcification was significantly affected by treatment conditions (F7,125 = 10.69, p < 0.0001 for zooxanthellate and F6,38 = 3.99, p = 0.0034 for bleached microcolonies, Figs. 2A and 2C). Exposure to light or oxygen significantly (p < 0.05) increased calcification of zooxanthellate microcolonies relative to dark controls while 1% glycerol with oxygen led to a significant decline in dark calcification (Fig. 2A). Treatment with glucose and oxygen or lower concentrations of glycerol with oxygen was not associated with a significant change in dark calcification for zooxanthellate microcolonies. In contrast, for bleached microcolonies light or oxygen alone had no effect but treatment with oxygen combined with fixed carbon increased dark calcification, with a significant increase (p = 0.0035) for the 1% glycerol plus oxygen treatment (Fig. 2C).

Figure 2 Relative calcification and respiration rates.

Relative calcification (A, C) and respiration (B, D) rates for zooxanthellate (A, B) and bleached (C, D) microcolonies maintained under each treatment condition. Treatments were: cont, control; ab, antibiotics; O2, ∼2× atmospheric pO2; glu20µm, ∼20 µM glucose with oxygen; glu1mm, ∼1 mM glucose with oxygen; gly1mm, ∼1 mM glycerol with oxygen; gly120mM, 120 mM glycerol with oxygen; light, light. With the exception of the light treatment, all incubations were conducted in the dark. N = 4 or greater for all treatments; symbols are mean; error bars are standard deviation; ∗, treatment significantly different from control at p = 0.05.

Effects of treatments on respiration

Respiration rates were significantly affected by treatment conditions (F6,65 = 26.96, p < 0.0001 for zooxanthellate and F6,43 = 23.48, p < 0.0001 for bleached microcolonies, Figs. 2B and 2D). All treatments (except antibiotics) were associated with higher oxygen consumption in the dark (p < 0.05) for zooxanthellate microcolonies with the largest increases for corals treated with oxygen and fixed carbon (Fig. 2B) (incubation in the light led to oxygen production, data were not compared with dark incubations). Similarly in bleached microcolonies treatment with fixed carbon combined with oxygen had the largest effect (p < 0.001) on respiration (Fig. 2D) (light had no effect on respiration in bleached colonies).

Effect of treatments on ATP

Treatments significantly affected ATP levels in zooxanthellate microcolonies regardless of the normalization procedure (F6,32 = 3.18, p = 0.0146 for normalization to protein, F6,32 = 6.41, p = 0.0002 for dry weight, F6,32 = 5.71, p = 0.0004 for surface area), with reduced ATP levels being associated with glycerol (Fig. 3). For bleached microcolonies similar trends were observed, however only for normalization to surface area was the 1% glycerol with oxygen treatment significantly lower than control (F5,26 = 3.84, p = 0.0097, pairwise 1% glycerol to control p = 0.011).

Figure 3 ATP measurements.

ATP contents (μg) normalized to: mg protein (up triangle, plotted against left axis), cm2 surface area (filled circles, plotted against left axis), and g skeletal dry weight (squares, plotted against the right axis) for zooxanthellate (A) and bleached (B) microcolonies exposed to each treatment (per Fig. 2). N = 2 or greater for all treatments; symbols are mean; error bars are standard deviation; ∗, treatment significantly different from control at p = 0.05.

Discussion

Stirring

Flow limitation may in part explain the wide range of estimates for ‘light enhanced calcification’ (Gattuso, Allemand & Frankignoulle, 1999; Al-Horani, Tambutté & Allemand, 2007). In the initial experiments with different sized stir bars (Fig. 1) light calcification and oxygen production remained nearly constant, while smaller stir bars (and consequent reduced flow) were associated with declines in dark respiration and calcification. If such trends extend to still lower flow regimes, as may be suggested by the larger declines in dark calcification observed by Dennison & Barnes (1988) under unstirred conditions, large differences in light v dark calcification could be generated by incubating corals in static or poorly stirred containers. As suggested by experiments with elevated oxygen (Fig. 2), and experiments under aerated versus un-aerated conditions (Rinkevich & Loya, 1984), oxygen limitation may be the primary factor responsible for reduced calcification in the dark. The effects of oxygen limitation are likely to be amplified under low flow conditions due to the increased thickness of the diffusive boundary layer (e.g., Kuhl et al., 1995), as has been established for respiration (e.g., Shick, 1990).

Calcification

For corals with intact symbiosis (zooxanthellate microcolonies), oxygen appears to be the primary factor limiting dark calcification (Fig. 2A). A similar rate enhancement was seen with both light and oxygen in zooxanthellate microcolonies, suggesting oxygen production by the symbionts is the primary mechanism for light enhanced calcification. Similar observations have been made previously (Colombo-Pallotta, Rodríguez-Román & Iglesias-Prieto, 2010; Wijgerde et al., 2012), however earlier studies have not monitored pCO2 when adjusting pO2, thus the possibility of changes in saturation state being responsible for the reported increase in calcification could not be ruled out. Here pCO2 has been controlled during oxygen manipulation to ensure that the observed effect is a result of oxygen elevation and not concomitant degassing of CO2. In contrast to Wijgerde et al. (2012), we found no indication that oxygen tensions of ∼200% (∼400 µmol/kg sw) negatively affected coral growth. Indeed it seems surprising that such a modest increase in external pO2 would be associated with substantial declines in calcification given that higher pO2 is present at the surface of the coral tissue when photosynthesis is occurring (Shashar, Cohen & Loya, 1993), and that such conditions are usually associated with enhanced calcification e.g., light enhanced calcification.

Oxygen alone is not sufficient to increase calcification, as seen in bleached microcolonies (Fig. 2C). A respiratory substrate must also be present to allow the coral to benefit from increased oxygen tension. The precise nature of the respiratory substrate appears to be of less importance than its concentration as suggested by an increase in calcification for both glucose and glycerol which appears to be enhanced at high substrate concentrations. However, high concentrations of fixed carbon in the seawater are not necessarily beneficial to calcification as glycerol concentrations which enhanced calcification in bleached microcolonies reduced dark calcification in zooxanthellate microcolonies (Fig. 2; Colombo-Pallotta, Rodríguez-Román & Iglesias-Prieto, 2010).

The similarity of the results of the current study and those of Colombo-Pallotta, Rodríguez-Román & Iglesias-Prieto (2010) suggest that oxygen limitation in the dark for zooxanthellate corals is a common phenomenon, affecting both branching (S. pistillata, current study), and massive corals (Montastraea faveolata, Colombo-Pallotta, Rodríguez-Román & Iglesias-Prieto, 2010). However, although oxygen may generally be limiting calcification at night in zooxanthellate corals, there is likely a gradual depletion in respiratory substrates over the course of the night and eventually respiratory substrates will become limiting even for zooxanthellate corals. This may help to explain why declines in calcification with prolonged dark incubations have been observed in some studies (Al-Horani, Tambutté & Allemand, 2007) but not others (Moya et al., 2006), and be partly responsible for the lack of an enhancement of calcification in earlier studies of fixed carbon supplementation (Vandermeulen & Muscatine, 1974; Chalker, 1975). For corals in the natural environment, there is likely to be a high degree of variability amongst corals as to whether fixed carbon or oxygen is limiting in the dark, depending upon such factors as flow, light history, photosynthetic rates, etc.

Respiration

Respiration was similarly affected by oxygen and fixed carbon for both zooxanthellate and bleached microcolonies (Figs. 2B and 2D), with increased rates at elevated pO2 and further increases with added fixed carbon. Despite increases in respiration, calcification did not necessarily increase, with different patterns being observed in bleached versus zooxanthellate microcolonies (Fig. 2). Since treatments had different effects on calcification but similar effects on respiration for zooxanthellate v bleached microcolonies, there could be fundamentally different processes occurring e.g., additional fixed carbon stressing the carbon replete zooxanthellate microcolonies, yet benefiting the fixed carbon limited bleached colonies, or use of additional fixed carbon (and energy) in biosynthetic pathways for zooxanthellate microcolonies and respiration for bleached colonies. However it is also possible that the additional fixed carbon is having similar effects for both zooxanthellate and bleached corals. Since the oral tissue layers (which do not carry out calcification) are most exposed to the surrounding environment, they are likely to be more affected by changes in the composition of the external seawater. So oxygen and fixed carbon in the seawater are likely to have a greater effect on the oral tissue layers than the aboral tissue layers, thus the changes in respiration may reflect primarily the oral tissue layers. For zooxanthellate microcolonies with their high respiration rates (Table S2) and greater tissue thicknesses (assumed from a higher protein content per unit surface area in zooxanthellate microcolonies relative to bleached (Table S2), which may be associated with a greater diffusion distance through the tissue), increases in respiration in the oral tissue layers in response to fixed carbon may reduce oxygen availability to the calicoblasts and thus reduce calcification (alternatively the increased CO2 produced by respiration could generate similar patterns). In bleached microcolonies, respiration rates are lower and the tissue thickness (protein content) is reduced, thus the increase in respiration by oral tissue layers may be insufficient to restrict the oxygen supply to the calicoblasts, thus allowing the calicoblasts to benefit from the supply of fixed carbon. Zooxanthellae are likely to be more effective than external substrates in enhancing calcification due to the production of oxygen and fixed carbon within the coral tissue, and thus closer to the calicoblasts. Though even photosynthesis may not be sufficient to completely eliminate hypoxia in parts of the calicodermis, as suggested by low pO2 values measured within some corals (Agostini et al., 2011), so hypoxia may still affect calcification even under light conditions.

Estimates of gross photosynthesis which have been made by combining net oxygen release in the light and oxygen consumption in the dark have long been known to be underestimates (Edmunds & Davies, 1988; Shick, 1990). Our results suggest that light respiration is stimulated by both oxygen and fixed carbon produced by photosynthesis, and, based on our observed stimulation, respiration rates are likely to be at least 20% higher in the light than in the dark. Even greater enhancements have been reported for short-term respiration measurements made immediately after light incubations (Edmunds & Davies, 1988; Colombo-Pallotta, Rodríguez-Román & Iglesias-Prieto, 2010). However, much of this enhancement of respiration may be due to respiration by the zooxanthellae, with a relatively small enhancement in the animal host (Agostini et al., 2013).

ATP

Despite different fixed carbon sources having similar effects on respiration and calcification, glucose and glycerol had markedly different effects on ATP (Fig. 3). Glycerol was associated with reduced ATP levels while little change was observed for other treatment conditions. Although it is not clear why such a difference is observed, it could be associated with high concentrations of glycerol stressing the corals as reduced ATP levels have been associated with stress events (Fang, Chen & Chen, 1991). Or it may reflect different pathways being used for glycerol versus glucose (e.g., Whitehead & Douglas, 2003), and the demands of those respective pathways on cellular ATP. Light v dark conditions appear to make relatively little difference in ATP levels—here we measured slightly lower values in the light, while previously slightly higher levels have been measured in the light (Al-Horani, Al-Moghrabi & de Beer, 2003), in neither case were the differences statistically significant, suggesting that corals generally regulate metabolic activities to maintain stable ATP levels, though glycerol may perturb this balance (Fig. 3). ATP levels have been suggested to be potentially useful indicators of coral health (Fang, Chen & Chen, 1991), however our results suggest that such an interpretation is far from clear—calcification, which is also used as an indicator of coral health, showed little relationship to ATP levels. However, ATP appears to be a potential indicator of coral biomass. ATP normalized to protein was similar for zooxanthellate and bleached microcolonies, but when normalized to either skeletal mass or surface area, ATP was lower in bleached microcolonies, suggesting that ATP is closely linked to biomass as has been observed in deep-sea corals (Hamoutene et al., 2008), and since tissue biomass is a potential indicator of coral health, ATP may be similarly useful.

Due to the nature of the measurements, we cannot, however, rule out a role for ATP levels in controlling calcification rates. Our measurements estimate total ATP and not the ATP content of the calicoblastic cells, nor the turn-over of ATP within the calicoblasts. Thus though the ATP content of the full tissue layer does not track calcification, we cannot assess whether ATP levels within different cell layers respond differently to treatments and what the ATP content of the calicoblasts may be.

Environmental Context

Although aquarium conditions are far removed from natural environmental conditions in many respects—with corals often experiencing constant temperatures, flow regimes, and light intensities under aquarium conditions, many of the parameters measured agree well with measurements made on corals taken from natural environments. ATP levels for a variety of corals collected from the wild were reported to range from 8.2 to 52.6 µg/g(Fang, Chen & Chen, 1991), our measurements (∼9.3–46.8 µg/g) fall well within this range, however Fang, Chen & Chen (1991) did not report values for S. pistillata. Our respiration rate of 12.5 µmol O2/d/cm2 (for zooxanthellate control corals) falls within the range measured in S. pistillata in the field: 0.5–16.6 µmol O2/d/cm2 (Mass et al., 2007). Calcification rates (23.7 µmol CaCO3/d/cm2 for zooxanthellate control corals), were however higher than recently reported in the field for S. pistillata—0.24–9.6 µmol CaCO3/d/cm2 (Mass et al., 2007). However, in other species, higher calcification rates have been reported during the light: 24 µmol CaCO3/d/cm2 (Schneider et al., 2009), 88 µmol CaCO3/d/mg protein (Dennison & Barnes, 1988), and in the dark: 44 µmol CaCO3/d/mg protein (Dennison & Barnes, 1988) versus our 10.2 µmol CaCO3/d/mg protein. Despite the differences between aquarium and natural conditions, for the range of physiological parameters measured, values fall within the range encountered in the natural environment. Thus, results from the current laboratory experiment likely apply to the natural environment, though with the diversity of species, growth forms, and environmental conditions found in nature, it is difficult to assess how common oxygen or fixed carbon limitation may be in corals growing under natural conditions.

Conclusions

Oxygen limitation plays a role in explaining reduced calcification rates during dark periods for scleractinian corals. Factors such as flow rate and tissue structure likely contribute to the degree of oxygen limitation and thus influence ‘light enhanced calcification’. Fixed carbon can also be limiting, as was the case for bleached microcolonies. However changes in total ATP in the tissue did not reflect changes in calcification, but rather changed in response to the form of fixed carbon. This suggests that any regulation of calcification by energy availability occurs via a pathway which is not responding directly to the total ATP level in the tissue.

Supplemental Information

Supplemental Information Flow speeds, ATP extraction comparison, Average control values, Incubation chamber illustration

Supplemental material containing additional methods for estimating flow speeds, a comparison of ATP extraction protocols, summary tables for ATP extractions and for control values, and an additional figure to illustrate the incubation chambers.

Click here for additional data file.

We would like to thank our colleagues for assistance and fruitful discussions, particularly R Grover, S Sikorski, C Rottier, C Godinot, M Naumann, N Segonds, D Desgre, A Venn, N Techer, C Ferrier-Pagès, and E Elia. We also thank G Gaetani (Woods Hole Oceanographic Institution) and M McCulloch (University of Western Australia) for the use of equipment.

Additional Information and Declarations

Competing Interests

Author Contributions

The authors declare there are no competing interests. Eric Tambutté, Denis Allemand, and Sylvie Tambutté are employees of the Centre Scientifique de Monaco.

Michael Holcomb conceived and designed the experiments, performed the experiments, analyzed the data, contributed reagents/materials/analysis tools, wrote the paper, prepared figures and/or tables, reviewed drafts of the paper.

Eric Tambutté, Denis Allemand and Sylvie Tambutté conceived and designed the experiments, contributed reagents/materials/analysis tools, wrote the paper, reviewed drafts of the paper.

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
