# Peer review of "Light enhanced calcification in Stylophora pistillata: effects of glucose, glycerol and oxygen"

_PeerJ, doi:10.7717/peerj.375_

## Round 0.1 · original submission · Minor Revisions

This is carefully done contribution and all reviewers agreed that should be accepted for publication. I recommend you address all their comments that will help you improve it. In particular, the issue of testing for residual zooxanthellae presence in the bleached samples is important to address, as well as expanding the discussion on flow raised by one reviewer.

Nice piece of work. Congratulations!

Reviewer 1 ·

Basic reporting

The article adheres to PeerJ principles and provides novel insights. The authors do a good job of describing the background and the potential pitfalls of the experimental design to measure the respiration /calcification rate and the stirring procedure. The authors report on the findings that oxygen availability does have an effect on the rate of calcification yet they also add that the total ATP concentration does not explain the rate of calcification in Stylophora pistillata. The text overall is readable and provides pertinent results to the coral community.

Experimental design

The authors discuss thoroughly the methods used, and employ the needed tools to answer the questions they ask. They us well established techniques to measure ATP and seawater chemistry.

Validity of the findings

The findings are valid, and provide a reasonable contribution. The results are in concordance with similar studies.

Additional comments

The authors cite this paper colombo-pallota and colombo-pallotta differently while they are the same, there seems to be a typo in the text multiple times with this reference. on line 194 and 332 333 336 387

Reviewer 2 ·

Basic reporting

No Comments

Experimental design

No comments

Validity of the findings

No comments

Additional comments

This interesting manuscript explores various parameters affecting light enhanced calcification in zooxanthellate and bleached corals. It is well established that corals calcify faster in the light then in the dark, so understanding the light enhanced mechanism would be of wide interest. However, I see some issues that need addressing before I feel that this work is ready, including inclusion of more data and some rewriting.

General comments:
1. It is necessary to show that the bleached colonies don’t contain zooxanthellae and don’t photosynthesize.
2. The first parameter measured is the effect of stirring on physiological parameters. The authors should add a paragraph in the introduction explaining the effect of flow on coral physiology; explain boundary layer effects and so on. In addition, a short experiment showing the effect of the different stirring should be added. The effect of the stirring can be visualized either by gypsum desolation or fluorescein dye. In the first paragraph of the discussion, boundary layer thickness can also explain the oxygen limitation in the dark (see fig. 3 at Kuhl et al 1995_ Mar. Ecol. Prog. Ser 117: 159-172, 1995).
3. Several statements lack references, e.g., line 30-32; add reference(s) for growth rate of azooxanthellate coral and beached coral.
4. Please add a brief description of the method used for calculating oxygen concentration.
5. Given the fact that the tissue thickness in zooxanthellate colonies is greater than in bleached colonies, the gradient of oxygen concentration in the tissue will be different due to the molecular diffusion coefficient. An assessment of the above difference in mass fluxes can be calculated. Therefore, reduced oxygen concentration in the calicoblastic layer of the zooxanthellate colonies could be the result of the influx gradient and not only the respiration.

Reviewer 3 ·

Basic reporting

This study addresses the effects oxygen and a veriety of dissolved organic carbon sources on the metabolism of microcolonies of Stylophora hanging on a thread in a conical flask. Overall, the study is well written in clear English and does a good job at presenting the context, approach, methods, results and interpretations

Experimental design

The experimental design is simple one way ANOVA in which the various treatments are compared. Each treatment consists of a series of conical flasks in which corals hang in seawater and are exposed to different conditions. This is a classic physiologists view of how corals function, and this is pretty much as far from a coral reef context as these kinds of studies can be. The work has been carried out with considerable attention to detail and the discoveries make an important contribution to the field. I do have some reservations on how far the results can be extrapolated (see below).

Is there potential for the glycerol and glycine additions to have osmotic effects?

I have three comments regarding the design.

1. Hanging corals in flasks with spin bars is characteristic of some of the earliest studies in the coral physiology that were conducted long before the importance of flow was appreciated. With the constraints of their system, the authors make a realistic attempt to address this issue by using spin bars of varying size and plotting effect against spin-bar size. The inference that flow speed has an effect and therefore a certain size bar is most useful it valid. However, It is difficult to translate the text into a picture of what the design looked like – I picture a coral sitting in the eye of a vortex created by the spin bar, and thus the actual flow is difficult to evaluate. It would be most helpful if the authors could present data on the rough cm/s experienced by their corals. Given the expansion on this topic in the discussion, presentation of flow speeds would provide a greater appreciation of ecological relevance.

2. The paper would be improved by presenting the main ATP method used in the body of the paper. The other comparative approaches can stay in the supplemental material.

3. The presentation of dependent variables as relative rates (Figs. 1, and 2) is unhelpful and will limit the impact of this research. Relative values can hide such a plethora of variance in the numerator and denominator and make it impossible to know whether the absolute values of the dependent variables are meaningful. I strongly recommend these graphs be replotted showing the actual units of measurement.

Validity of the findings

Overall, the findings are interesting and valid. They should stimulate a lot of further research.

However, I feel it is important to recognize the strongly lab/physiological nature of these experiments and the challenges of extrapolating the results to the field. Key issues are:

• the colonies are exceptionally small and long removed from a coral reef
• the flow regime is unspecified and unlike flow speeds found on reefs
• the light regime is exceptionally low compared to that occurring in shallow water (≤ 5 m depth)
• the incubations are very short

I firmly believe that studies such as this one are important and have critical role in the global efforts to understand how corals function and the ways in which coral reefs will respond to climate change. However, it is important to recognize both the strengths and limitations of these efforts. The present manuscript does not go far enough to objectively recognize the limitations of the study. Table S2 is most helpful in this regard, because it presents the data necessary to evaluate whether the values recorded are close to what might be expected for corals on a reef. I did a few quick calculations for myself, and some variables seem close to what I would expect (respiration), while others (e.g., calcification) are high. What about ATP levels? (etc.). It would be most helpful to add a paragraph that places the results in a broader comparative context relative to measurements of similar things in other corals collected fresh from coral reefs.

Additional comments

This is a nice study that should be published. I have a few concerns but these should be easy to address.

---

## Round 0.2 · accepted · Accept

Thank your for your thorough revision.